# Rapid Identification by Resequencing-Based QTL Mapping of a Novel Allele *RGA1-FH* Decreasing Grain Length in a Rice Restorer Line ‘Fuhui212’

**DOI:** 10.3390/ijms241310746

**Published:** 2023-06-28

**Authors:** Shiying Ma, Yifan Zhong, Shuyi Zheng, Ying He, Sihai Yang, Long Wang, Milton Brian Traw, Qijun Zhang, Xiaohui Zhang

**Affiliations:** 1State Key Laboratory of Pharmaceutical Biotechnology, School of Life Sciences, Nanjing University, Nanjing 210023, China; mg1630053@smail.nju.edu.cn (S.M.); zhongyf056@163.com (Y.Z.); njzhengsy@163.com (S.Z.); njunxuheying@163.com (Y.H.); sihaiyang@nju.edu.cn (S.Y.); wanglong@nju.edu.cn (L.W.); traw9@hotmail.com (M.B.T.); 2Institute of Food Crops, Jiangsu Academy of Agricultural Sciences, Nanjing 210014, China

**Keywords:** Fuhui212, grain length, grain size, QTL mapping, RGA1, rice

## Abstract

Grain size is one of the most frequently selected traits during domestication and modern breeding. The continued discovery and characterization of new genes and alleles in controlling grain size are important in safeguarding the food supply for the world’s growing population. Previously, a small grain size was observed in a rice restorer line ‘Fuhui212’, while the underlying genetic factors controlling this trait were unknown. In this study, by combining QTL mapping, variant effect prediction, and complementation experiments, we recovered a novel allele *RGA1-FH* that explains most of the phenotypic changes. The *RGA1-FH* allele contains an A-to-T splicing site variant that disrupts the normal function of RGA1. While population analysis suggests extremely strong artificial selection in maintaining a functional allele of RGA1, our study is the first, to the best of our knowledge, to prove that a dysfunctional RGA1 allele can also be beneficial in real agricultural production. Future breeding programs would benefit from paying more attention to the rational utilization of those overlooked ‘unfavored’ alleles.

## 1. Introduction

Rice is the most important cereal crop in the world. Grain characteristics, such as length, width, thickness, and number, are important agronomic traits directly contributing to rice yield and quality. Genes affecting grain size are, therefore, major selection targets in rice breeding. To date, many genes have been isolated and characterized for their functional roles in affecting grain size, including transcription factors, ligases, hydrolases, and phosphatases (reviewed in refs. [1,2]). For example, SQUAMOSA PROMOTER-BINDING PROTEIN-domain (SBP) transcription factors, such as GLW7 and GW8, regulate grain length and grain width, respectively [3,4]. *GW2* encodes an RING-type E3 ubiquitin ligase and controls rice grain width and weight [5].

G proteins (guanine nucleotide-binding proteins) are also among the most important in regulating grain size, including the Gα protein (RGA1), Gβ protein (RGB1), and Gɣ protein (RGG1, RGG2, GS3, DEP1, and GGC2) [6]. The rice *rga1* mutant, also called *d1* mutant, exhibits dwarf traits and small round seeds [7,8]. The overexpression or suppression of RGB1 reduces grain size [6,9,10]. The five Gɣs genes in rice have different roles in regulating grain size. DEP1 and GGC2 both promote grain growth, whereas RGG1, RGG2, and GS3 repress grain growth, in apparent relation to having long vs. short cysteine-rich C-terminal ‘tail’ domains, respectively [6,10].

Diverse *rga1* alleles have been identified in rice mutants. Katsuyuki Oki et al. identified ten *rga1* alleles from different donors, including various mutation types: (1) base substitution, (2) 80 bp deletions, 19 and 65 bp insertions [11]. Nine of ten alleles of *rga1* exhibited a more severe phenotype than the wild-type allele, whereas another showed a mild phenotype. All the mutants had small, round seeds. A low abundance of *RGA1* transcripts in the cultivar Xueheaizao showed a semi-dwarf phenotype [12], perhaps due to a sequence polymorphism identified in the *RGA1* promoter. Additionally, an epigenetic *rga1* allele has been reported, named *Epi-d1* [13], which is associated with DNA methylation and repressive histone marks, resulting in an *Epi-d1* mutant that is a dwarf with small round grains.

‘Fuhui212’ is a restorer line derived from an irradiation mutant with small grains, a dwarf stature, and an erect panicle [14] that has been identified as a promising background for investigating the genetic control of grain size. Using QTL-Seq, a prior study mapped a candidate QTL, *qTGW5.3*, which was associated with variations in the grain length and weight of the progeny population derived from a ‘Hui12-29/Fuhui212’ cross [15]. However, because the candidate QTL region spans from 15 to 20 Mb on Chromosome 5, a region containing hundreds of genes, the causal gene(s) have not been identified previously.

In this study, we analyzed the F_2_ populations derived from a cross between ‘Fuhui212’ and a popular cytoplasmic male sterile (CMS) line, ‘Quan9311A’ [16]. Combining QTL mapping and allelic variation analyses, we were able to map the candidate gene to the *RGA1* locus. Plants with a small grain size have a splice acceptor variant which causes the disruption of normal coding frames, leading to the dysfunction of *RGA1*. The complementation of *rga1* plants with normal *RGA1* restored the phenotypes to a larger seed size, confirming that its role is responsible for the seed size control in the analyzed cross. The presence of this ‘unfavored’ version of *RGA1* in an excellent restorer line, therefore suggesting the importance of preserving the malfunctional alleles of functional important genes.

## 2. Results

### 2.1. Phenotyping Analysis of Grain Size in the Cross of ‘Fuhui212’ and ‘Quan9311A’

To decipher the gene(s) responsible for the observed small-grain phenotype in ‘Fuhui212’, we obtained F_1_ progeny from a cross between ‘Fuhui212’ and ‘Quan9311A’, a larger-grained variety. The F_1_ progeny yielded grain sizes equivalent to ‘Quan9311A’, suggesting the small grain size as a recessive trait. We further obtained F_2_ samples through the selfing of F_1_ samples. Consistent with a single dominant locus, the segregation ratio of plants with large and small grains was roughly 3:1 in F_2_ samples (188 plants measured with 146 large and 42 small, Chi-squared = 0.7092, *df* = 1, *p* = 0.3997). The phenotyping results (Figure 1a–e and Appendix A) revealed that the larger grain size is mainly caused by an increased grain length (Figure 1d, Welch’s *t*-test, *p* = 8.03 × 10^−40^), while the grain width became slightly wider in small grains than in larger grains (Figure 1e), consistent with previous reports [15].

### 2.2. Resequencing and Genotyping the F_2_ Generation Individuals

From the segregating F_2_ population, we picked 30 plants with a large grain and 19 plants with a small grain for whole-genome sequencing (WGS). For each plant, we obtained 16.1 Gb clean data, corresponding to an average depth of 43-fold (Appendix A). The sequencing reads of one parental sample, ‘Quan9311A’, were also collected from the NCBI SRA database (accession ID: SRR12523731) and were included in downstream analysis. After mapping to the Nipponbare reference genome, we called ~8 M single-nucleotide-polymorphisms ‘SNPs’ and ~0.7 M insertion/deletions ‘INDELs’ across the 50 (49 F_2_s + 1 parent) analyzed samples.

### 2.3. QTL Mapping Identifies the Strongest Signals in the Regions of Chromosome 5

Prior to QTL mapping, we first screened the 8.7 M variant sites to obtain informative markers, i.e., differences between the two cross parents. We removed the non-informative and non-segregating variants (mostly owing to *indica-japonica* differences, as we used Nipponbare as the reference) as well as variants with poor quality (see Methods for details), yielding 935,780 filtered polymorphic markers (Appendix A). We phased the genotypes of each marker based on the genotype of ‘Quan9311A’. For each marker, the parental source of every F_2_ sample (i.e., homozygous in Quan9311A-type, heterozygous, or homozygous in Fuhui212-type) is assigned by comparing with the genotype of ‘Quan9311A’. This dense marker panel allowed us to reach a high resolution in mapping candidate causal regions.

We used R/qtl [17] to perform QTL mapping in a single-QTL model. The distribution of LOD (logarithm of odds) scores revealed one candidate region with the highest LOD values in chromosome 5 from 5 Mb to 18 Mb (Figure 1f), residing at the similar region identified before [15]. Much weaker signals were found in several other narrow regions (Appendix A). The investigation of genes in different regions suggested that, unlike the region in Chromosome 5, nearly all other regions were enriched with (retro)transposable elements (TEs, Appendix A) and are therefore more likely to represent noise. The large size of the region in chromosome 5 was expected, given the limited amount of crossover within a single generation. Taken together, the identified QTL region in chromosome 5 was the only possible region for harboring the dominant locus.

### 2.4. Identification of a Causal Variant and Target Gene

The ~13 Mb candidate region in chromosome 5 overlapped with 2024 genes, of which 1140 were non-TE protein-encoding genes (Appendix A). To identify the most likely causal gene, we first searched for variant sites within this region that strictly co-segregated with the grain length. Since a small grain size is a recessive trait, all plants with shorter grain lengths should have the same homozygous genotype as ‘Fuhui212’, while other plants with longer grain lengths could be either heterozygous or Quan9311A-type homozygous. We found 4851 variants that precisely followed this segregation pattern within the 13 Mb region (Appendix A). We also predicted the effect of each variant based on the gene annotations of the reference genome to see whether any variant might cause functional alterations in these protein-coding genes. Over 95.9% of the co-segregated variants were predicted to have no large impact on coding regions (including variants outside CDS and synonymous variants), and ~3.5% only change or delete/add a few codons (Appendix A). Only 24 high-impact variants were found, including 15 causing frameshift changes, 6 introducing premature stop codons, and 3 causing a loss of start codons, stop codons, or splice acceptors (Appendix A). These 24 high-impact variants were hence considered as the most possible causal variants.

There were 14 genes affected by these 24 high-impact variants, of which 6 genes were affected by 2 or more high-impact variants (Appendix A). Most of these variants (20/24) were associated with the ‘Quan9311A’ genotype, meaning the functional copy of these genes would lead to a small grain size if any of them was indeed responsible for the phenotypic changes, i.e., the causal gene acting as a negative regulator. In contrast, the remaining four variants that were associated with the ‘Fuhui212’ genotype would suggest the opposite, i.e., a small grain size caused by the disruption of normal function and the causal gene acting as a positive regulator. Since it is clear that the change in the grain length happened in ‘Fuhui212’ (a shortened length caused by gamma-radiation) rather than in ‘Quan9311A’, we know that the four Fuihui212-associated variants were the most possible causal variants, and the causal gene is most likely a positive regulator. Since the genes affected by two of the four Fuhui212-associated variants were also affected by Quan9311A-associated variants (Appendix A), these two could also be ruled out, given that the affected genes should be deficient in both ‘Fuhui212’ and ‘Quan9311A’. Therefore, only two genes, *LOC_Os05g26890* and *LOC_Os05g26980*, were the possible causal genes of an altered rice grain size.

Further inspection of the functional annotations of all 14 genes (Appendix A) suggested most of them, including *LOC_Os05g26980*, showed no direct link to grain traits or even had no annotated functions. Only one gene, *LOC_Os05g26890/Os05g0333200*, known as *Dwarf1* (*D1*) or *G-protein alpha subunit 1* (*RGA1*), was found to be directly responsible for a small grain trait when disrupted [8,18,19]. The *RGA1* allele derived from ‘Quan9311A’ encodes a normal α-subunit of the GTP-binding protein, while the Fuhui212-derived allele (designated as *RGA1-FH*) harbors an A-to-T mutation in the splice acceptor site of the fourth intron, leading to the premature termination of the subsequent coding frame (Figure 1g and Appendix A). Confirmation through cDNA sequencing revealed that this splice site mutation caused a 13 bp deletion in the transcript compared to the wild-type, resulting in a premature termination codon in the subsequent coding frame of the derived ‘Fuhui212’ allele. We then analyzed the genotype distribution of *RGA1* in our sequenced F_2_ population (Appendix A). Our results show that all F_2_ individuals with a small grain have homozygous A-to-T splicing site mutations in *RGA1*, while large-grain individuals have both homozygous wild-type and heterozygous A-to-T splicing site mutations in *RGA1*. Therefore, the *RGA1* locus has the highest potential to be the dominant causal locus of a decreased rice grain length.

### 2.5. Complementation Test Confirmed RGA1 as the Dominant Locus

To further confirm whether *RGA1* is responsible for the observed phenotypic changes, we performed a complementation experiment by transferring the Quan9311-derived normal *RGA1* allele into the plants carrying the homozygous *RGA1-FH* alleles (Figure 2a). We cloned the *RGA1* with its native promoter sequence from a large-grain plant F_2L_-30 which inherited homozygous *RGA1* alleles from ‘Quan9311A’. A small-grain plant F_2S_-16 with homozygous *RGA1-FH* was chosen as the receptor plant for a complementation test. A total of 11 transgenic seedlings were obtained, which conferred resistance to hygromycin, indicating the successful integration of the vector. We further performed q-PCR quantification to verify whether the transgenic vector was properly expressed after being transformed into the recipient plants. The premature termination codon introduced in the *RGA1-FH* allele predicts a low expression level in those plants with homozygous *RGA1-FH* owing to the Nonsense-mediated mRNA Decay (NMD) mechanism [20]. In line with this expectation, we observed that the expression level of *RGA1* in small-grain individuals with the *RGA1-FH* homozygous genotype is much lower than that in large-grain individuals with normal *RGA1* alleles (Appendix A). After transformation, the expression level of *RGA1* in transgenic plants was confirmed to be successfully restored, even to a higher level (Appendix A).

Subsequently, we measured the phenotypic changes, including the plant height, grain width, and grain length, prior to and after the complementation of *RGA1*. Our results showed that the transgenic plants were taller than the small-grain individuals, and there was no significant difference in grain length and grain width compared to the large-grain individuals (Figure 2b–d and Appendix A, complement versus large grain, Welch’s *t*-test, *p* = 0.867 for grain length, *p* = 0.740 for grain width). Therefore, the plant stature and grain size have been successfully restored in the complemented plants when compared to the plants with normal *RGA1* alleles. In summary, by complementation tests, we confirmed that the *RGA1* locus is the dominant locus, and the novel identified *RGA1-FH* allele is the causal allele that causes the small-grain trait with a shortened grain length in ‘Fuhui212’.

## 3. Discussion

To date, there are at least 13 *rga1* mutants that have been characterized in the literature [7,11,21,22,23] (Figure 2f). *RGA1* is considered to provide the baseline for the grain size regulation pathway composed of the three Gγ proteins [6], and knocking out *RGA1* leads to significantly smaller grains, which is clearly undesirable for breeding purposes. An investigation of the ~3000 Asian cultivated rice panel [24] reveals that all 3000 cultivars carry normal *RGA1* alleles, with only one amino acid change in the second-to-last exon present in a few indica cultivars (Figure 2e). The further investigation of RGA1 in eight wild rice species (see Methods) revealed that the intact RGA1 protein is present in nearly all investigated wild rice genomes. This suggests strong negative selection as well as artificial selection acting at the locus, resulting in a dearth of *RGA1* alleles among rice cultivars, which is in line with its fundamental importance in plant development and stress responses; additionally, its role serves as the basis for a large grain size.

Our finding that the *RGA1*-deficient allele, *RGA1-FH*, is responsible for the short grain of an excellent restorer line provides an intriguing case for how to incorporate mutant lines with several unfavored traits into real breeding programs. The application in hybrid rice not only shields the unfavored recessive alleles but also could benefit from other favored traits of the mutant lines. It is worth noting that, despite the small seed, ‘Fuhui212’ was characterized as having a good grain quality, excellent blast-resistance, strong restoration ability, and even less hampered yield [14]. Therefore, the application of this *RGA1*-deficient line is possibly not restricted to hybrid rice. Prior studies have showcased that the *rga1* mutants had a decreased susceptibility to water stress [25] as well as photoinhibitory damage [26] owing to dwarf and erect leaves, indicating the possibility of incorporating them for developing stress-tolerant rice. Taken together, these cases suggested that rice lines with dysfunctional trait-related genes such as *RGA1* should be more comprehensively evaluated before excluding them from real breeding programs. More work should be carried out to test both ‘favored’ and seemingly ‘unfavored’ alleles in guiding rice breeding and production.

## 4. Materials and Methods

### 4.1. Sampling and Whole-Genome Sequencing

The original cross between ‘Fuhui212’ and ‘Quan9311A’ was made by Jiangsu Academy of Agricultural Sciences (JAAS), Nanjing, China. Seeds of 3~5 F_1_ hybrid plants were harvested and planted in 2019 at Lishui, Nanjing, China. Seeds of F_2_ plants were planted in 2020 at Nanjing University, Nanjing, China. Fresh leaves of F_2_ plants were collected at the heading stage and stored at −20 °C. The grain-related traits (e.g., grain length, grain width, and 1000-grain weight) were measured for both F_2_ plants using Wanshen SC-G automatic seed analysis and a 1000-grain weight instrument (V2.1.2.4, Wseen Detection Technology Co., Ltd., Hangzhou, China).

The DNA of leaf samples collected from F_2_ was extracted using the CTAB method [27] and sent to BGI (Beijing Genome Institution, Beijing, China) for quality testing, library construction, and genome sequencing. The DNA was fragmented to 300 bp inserts and sequenced in the paired-ended mode with a 150 bp read length on the MGISEQ2000 platform. Resequencing reads for ‘Quan9311A’ were obtained from the NCBI SRA database (https://www.ncbi.nlm.nih.gov/sra/, accession id: SRR12523731, accessed on 18 October 2020). Information about RGA1 alleles in eight wild rice species was retrieved from ENA (https://www.ebi.ac.uk/ena, accessed on 19 June 2023.) under accessions GCA_000182155 (*Oryza barthii*), GCA_000231095 (*Oryza brachyantha*), GCA_000576495 (*Oryza glumipatula*), GCA_000789195 (*Oryza longistaminata*), GCA_000338895 (*Oryza meridionalis*), GCA_000576065 (*Oryza nivara*), GCA_000573905 (*Oryza punctata*), and GCA_000817225 (*Oryza rufipogon*).

### 4.2. Sequencing Data Process and Variants Calling

The sequenced reads were cleaned by trimming adapter sequences and removing low-quality reads (reads with >3% Ns or reads with >50% bases of a quality value < 20). Cleaned reads were mapped to the Nipponbare reference genome (Os-Nipponbare-Reference-IRGSP-1.0) [28] with BWA-mem (version 0.7.10-r789) [29]. The option ‘-M’ was used to mark shorter split hits as secondary for downstream compatibility. Read alignments were sorted, and the PCR duplicates were removed using Picard tools (https://broadinstitute.github.io/picard/, version 1.114, accessed on 18 October 2020). A further round of realignment around INDELs was performed with GATK (version 3.7) [30].

For variants calling, we ran GATK UnifiedGenotyper in the joint-calling mode with the option ‘-glm BOTH’ to call both SNPs and INDELs and also with the options ‘-rf MappingQuality-mmq 20’ to only consider reads with a mapping quality over 20 (i.e., expected mis-mapping probability lower than 1%). To obtain a qualified variant set, the raw variants were filtered by: (1) only retaining bi-allelic sites with a variant quality score no less than 50; (2) for each variant, requiring each sample to be supported by at least 10 reads and at most 90 reads (i.e., mean + 2 × SD); (3) since ‘Quan9311A’ is a homozygous inbred line, any heterozygous variants present in ‘Quan9311A’ would indicate putative mapping artifacts, so we removed variants within 150 bp of those ‘pseudo-heterozygous’ variants found in ‘Quan9311A’; (4) removing sites with over half of the samples missing.

### 4.3. QTL Mapping

The above filtered variants consisted of both those segregating between the two cross parents and variants between the two cross parents and the reference genome. The latter category is non-informative in QTL mapping, as the two cross parents have identical alleles and should be discarded. Since the full sequence of ‘Fuhui212’ was not available for this study, we excluded variants with a minor allele frequency < 5 to ensure the retained variants represented those segregating variants. The genotypes of all samples within each retained variant site were then compared to the genotype of ‘Quan9311’ to determine whether this sample is Quan9311-homozygous (homozygous genotype is the same as ‘Quan9311’), Fuhui212-homozygous (homozygous genotype differed from ‘Quan9311’), or heterozygous. These variant sites were used as variant markers for the subsequent QTL mapping process.

R/qtl [17] was used to perform QTL mapping with the above obtained variant markers as well as the recorded grain lengths. We first performed single-QTL analysis with all three methods implemented in R/qtl, namely, the EM algorithm, Haley–Knott regression, and multiple imputation. Since all three methods produced similar results, only the result of the Haley–Knott regression was reported for simplicity. LOD thresholds at a significance level of 0.05 and 0.01 were estimated by permutation tests with the options ‘method = hc, n.perm = 1000’. Finally, we calculated the interval estimate of the QTL location (i.e., the candidate QTL regions) as the 1.8 LOD support interval (the interval in which the LOD score is within 1.8 units of its maximum) [31].

We mapped the locations of genes within candidate QTL regions based on the annotation of the MSU Rice Genome Annotation Project (Release 7) [28]. The effects of variants were predicted using SNPeffect (version 5.0) [32].

### 4.4. cDNA Analysis of RGA1

Total RNA was extracted from the leaf tissues of ‘Quan9311A’, ‘Fuhui212’, and Nipponbare. These plants were grown for two weeks under long-day conditions (16 h:8 h, light:dark) at 28 °C, a relative humidity of 70%, and a light intensity of 15,000 Lux. The TaKaRa MiniBEST Plant RNA Extraction kit was used according to the manufacturer’s instructions. cDNA libraries were prepared using PrimeScript RT Master Mix (Takara, Kusatsu, Janpan). Primers (Appendix A) were used to amplify *RGA1* cDNA, and the PCR products were sequenced.

### 4.5. Genetic Complementation of RGA1

A 7063 bp genomic DNA fragment containing the entire *RGA1* coding region as well as the 2953 bp upstream region and the 470 bp downstream sequence of the ‘Quan9311A’ *RGA1* gene were amplified with Lamp Master Mix (Nanjing Vazyme Biotech Co., Ltd., Nanjing, China) (Appendix A) and cloned into the binary vector pCAMBIA1300. The recombinant plasmid was introduced into the ‘Fuhui212’ by *Agrobacterium tumefaciens*-mediated transformation. More than 10 independent T0 lines were obtained, and all were assayed by PCR. Positive transformants were collected and selfed to obtain the T_1_ progeny. Phenotypes were assessed in transgenic T_1_ plants. The expression of *RGA1* in transgenic lines was compared with large- and small-grain individuals by q-PCR to verify the successful expression of the transgenic vector in the receptor rice (Appendix A).

## Figures and Tables

**Figure 1 ijms-24-10746-f001:**
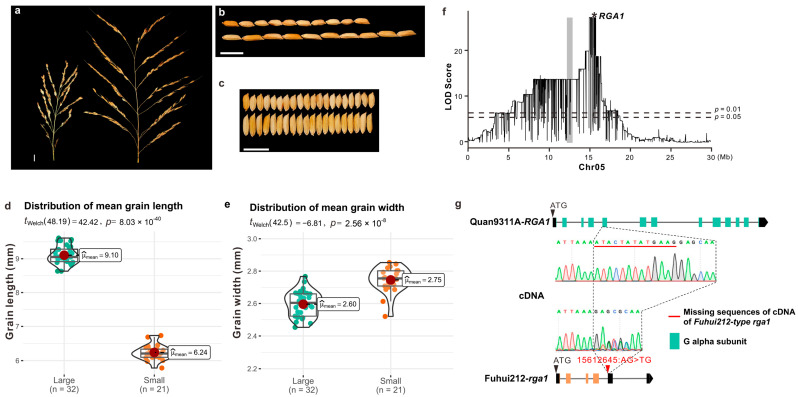
Mapping of a causal gene controlling the grain length in an experimental cross between ‘Quan9311A’ and ‘Fuhui212’. (**a**–**c**) Morphological differences in panicle type (**a**), grain length (**b**), and grain width (**c**) between the small-grain plants (**left** or **upper**) and large-grain plants (**right** or **lower**) from the F_2_ population. Scale bar, 1 cm. (**d**,**e**) Distributions of grain lengths (**d**) and grain widths (**e**) between small-grain plants (orange) and large-grain plants (green). Statistic tests using Welch’s *t*-test. (**f**) Distribution of LOD scores on chromosome 5. The asterisk indicates the position of *RGA1*. (**g**) Haplotype and cDNA sequence differences in *RGA1* between large- and small-grain individuals in the F_2_ population. The green squares indicate the functional G alpha subunit, and the orange squares indicate a nonfunctional subunit with premature termination. The red underline indicates the deleted sequence in the cDNA of *RGA1-FH*.

**Figure 2 ijms-24-10746-f002:**
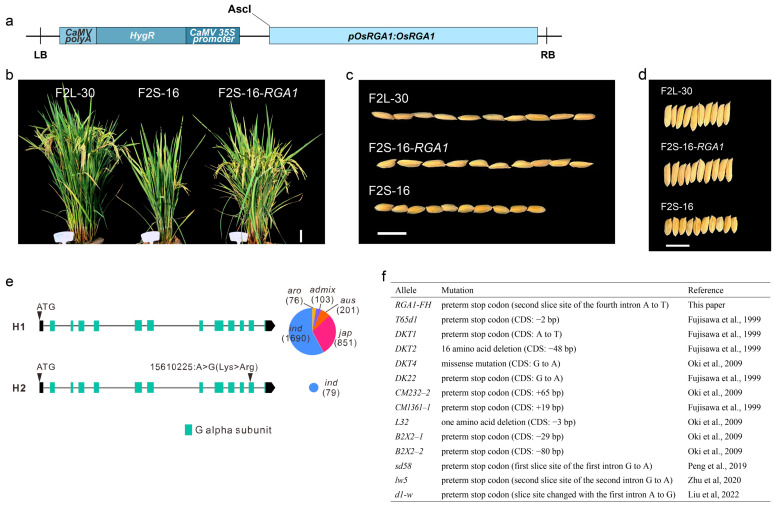
Complementation test recovers a shortened grain length and other related traits in the ‘Fuhui212’ background. (**a**) Schematic representation of the complementation vector (LB, left border; HygR., hygromycin resistance; RB, right border). (**b**–**d**) Morphological differences in plant stature (**b**), grain length (**c**), and grain width (**d**) among the Quan9311A-type *RGA1* of the F_4_ generation (F_2L_-30), Fuhui212-type *rga1* of the F_4_ generation (F_2S_-16), and the complemental lines (F_2S_-16-*RGA1*). Scale bars, 10 cm in (**b**) and 1 cm in (**c**,**d**). (**e**) Observed *RGA1* haplotypes and their frequencies in the rice 3000 database (pie chart). The green squares indicate the functional G alpha subunit. (**f**) Mutant alleles of *RGA1* reported to date [7,11,21,22,23].

## Data Availability

All the sequence reads used in this study have been deposited at NCBI SRA under the BioProject accession number PRJNA902968. The code for mapping, variants calling, and QTL mapping in this study has been stored at GitLab (https://git.nju.edu.cn/gattaca_bioinfo/grain_size, accessed on 30 May 2023).

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
