# Peer review of "Rapid Identification by Resequencing-Based QTL Mapping of a Novel Allele RGA1-FH Decreasing Grain Length in a Rice Restorer Line ‘Fuhui212’"

_ijms, 2023, doi:10.3390/ijms241310746_

Round 1

Reviewer 1 Report

This paper describes the cloning and characterization of an RGA-1-deficiency allele, RGA-FH, responsible for a short grain phenotype in rice. This work is well-described and very thorough. The figures are mostly clear and informative, although it’s necessary to zoom in quite a bit to read all of Figure 1e.

One comment/question about this discussion point:

“Investigation of the ~3,000 Asian cultivated rice panel [30] reveals that all 3,000 cultivars carry

normal RGA1 alleles, with only one amino acid change in the second to last exon present

in a few indica cultivars (Figure 2e). This suggests strong negative selection acting at the

locus, resulting in a dearth of RGA1 alleles among rice cultivars.”

This seems to suggest natural selection is selecting against the RGA1-FH allele.  If the 3,000 cultivars are from breeding programs, then have they not been subjected to strong human selection for large grain size?

Minor English language edits:

Line 180: “yielding” should be “yielded.”

Lines 195, 275: “casual” should be “causal.”

Author Response

We thank the reviewer very much for these constructive comments. These comments are all valuable and helpful for improving our manuscript. In light of these comments, we have substantially improved the English writing, and added more details to address the first question.

  1. “Investigation of the ~3,000 Asian cultivated rice panel [30] reveals that all 3,000 cultivars carry normal RGA1 alleles, with only one amino acid change in the second to last exon present in a few indica cultivars (Figure 2e). This suggests strong negative selection acting at the locus, resulting in a dearth of RGA1 alleles among rice cultivars.” This seems to suggest natural selection is selecting against the RGA1-FH allele. If the 3,000 cultivars are from breeding programs, then have they not been subjected to strong human selection for large grain size?

Response: Thank you for the comments. The reviewer is right that the malfunctional alleles of RGA1 are likely selecting against by natural selection, while the artificial selection also favors the normal alleles of RGA1 as the basis for large grain size. We added more evidences for this by further investigated the variants of RGA1 in eight wild rice genomes. Our results suggested that RGA1 is highly conserved even across different wild rice species, where the malfunctional allele is very rare. This is likely due to its large influence on plant development and stress responses. Subsequent artificial selection in breeding rice continues preserving the functional allele when select for large grain, as shown by the 3,000 rice cultivars. We have supplemented these details in the new manuscript.

  1. Comments and suggestions on English editing and writing

Response: Thanks for these suggestions. We have corrected all these issues according to the provided suggestions and comments. We also updated Figures 1d and 1e to make them larger and clearer.

Reviewer 2 Report

This is a well-conceived, executed, and reported study of the identity of the gene causing small grain size in rice. The manuscript goes further to point out how breeding values can be occluded by a superficial understanding of how genes are used in breeding programs. The paper makes excellent points about breeding strategies in addition to providing new insights into rice grain size determination and deserves to be published in IJMS.

Specific comments on the manuscript are provided below:

Abstract

Line 16) “…was unknown…” change to “…were unknown…”.

Line 17) “…RGA1-FH, which…” suggest change to “…RGA1-FH that…”.

Line 18) “…variant, which…” suggest change to “…variant that…”.

Line 21) “…malfunctional…” suggest change to “…dysfunctional…”.

Keywords: The order of terms should be in alphabetical order and formatting appears to be incorrect; consult with the journal for style preferences.

Introduction

Line 29) “…are thus major…” change to “…are, therefore, major…”.

Line 36) “G proteins…”. This term should be defined.

Line 39) “The five GÉ£s…” suggest change to “The five GÉ£ genes…”.

Line 43) “Katsuyuki Oki et al…” change to “Katsuyuki Oki et al.”.

Line 44) “…ten rga1 alleles…” change to “…ten rga1 alleles…”.

Line 44) “…types: 1 base…” change to “…types: (1) base…”.

Line 45) “…2-80 bp deletions,…” change to “…(2) 80 bp deletions,…”.

Line 46) “…exhibited a more severe phenotype, whereas another showed a mild phenotype.” More severe or mild than what? The wild type?

Lines 47-48) “…in a cultivar Xue-heaizao showed…” change to “…in the cultivar ‘Xue-heaizao’ showed…”.

Line 52 and elsewhere in the manuscript) “…Fuhui212 is a restorer line…” change to “…’Fuhui212’ is a restorer line…”.

Line 53) “…panicle [14], which…” suggest change to “…panicle [14] that…”.

Line 60) “…Quan9311A…” change to “…’Quan9311A’…”.

Line 61) “…to RGA1…” change to “…to the RGA1 locus…”.

Line 66) “… line hence suggested…” change to “…line, therefore, suggested…”.

Materials and Methods

Line 74) “…and were stored at -20 degree…” change to “…and stored at -20C…”.

Line 104) “…variants segregated…” suggest change to “…those segregating…”.

Line 106) “…latter is…” change to “…latter category is…”.

Line 121) “Final, we calculated…” change to “Finally, we calculated…”.

Lines 124-125) “To check the genes within candidate QTL regions, we mapped the gene location based on the annotation of MSU Rice Genome Annotation Project…” suggest change to “We mapped the locations of genes within candidate QTL regions based on the annotation of the MSU Rice Genome Annotation Project…”.

Line 129) “…which were grown for two weeks…”. How were these plants grown? Perhaps this is described elsewhere but should be specified since mRNA profiles will change with different growing methods.

Line 134) “…entire coding region…” change to “…entire rga-1 coding region…”.

Results

Line 151) “…was roughly 3:1 in F2 samples…”. The numerical results should be reported and a non-parametric statistical test (chi-square) of the 3:1 ratio confirmed.

Line 152) “…reveal that…” results should be reported in past tense “…revealed that…”.

Lines 172-174) “…we called ~8 M single-nucleotide-polymorphisms (SNPs) and ~0.7 M insertion/deletions (INDELs) across the 50 (49 F2s + 1 parent) analyzed samples. Suggest change to “…we called ~8 M single-nucleotide-polymorphisms “SNPs” and ~0.7 M insertion/deletions “INDELs” across the 50 (49 F2s + 1 parent) analyzed samples.

Line 180) “…which yielding 935,780 filtered…” change to “…yielding 935,780 filtered…”.

Lines 180-181) “We phased the genotypes of each marker based on the genotype of Quan9311A.” The reviewer is confused about what the authors mean by “phased”. More explanation would be helpful.

Lines 187-188) “…suggests that unlike the region in Chromosome 5,…” change to “…suggested that, unlike the region in chromosome 5,…”.

Line 195) “…likely casual…” change to “…likely causal…”.

Lines 196-197) “…for variant sites which strictly co-segregated with grain length within this region.” Suggest change to “…for variant sites within this region that strictly co-segregated with grain length.”.

Line 197) “Since small grain is a recessive trait, in the co-segregated locus all plants…” change to “Since small grain size is a recessive trait, all plants…”.

Line 200) “We found 4,851 variants exactly followed this segregation pattern…” change to “We found 4,851 variants that precisely followed this segregation pattern…”.

Lines 201-202) “Over 95.9% of the co-segregated variants were predicted to have no large impact on coding regions…” the reviewer is confused by this assertion. More explanation is needed; what impacts are inferred?

Line 217) “…we can know…” suggest change to “…we know…”.

Line 222) “…were the possible causal genes…” change to “…were the possible causal genes of altered rice grain size…”.

Line 229) “…of fourth…” change to “…of the fourth…”.

Line 238) “…dominant causal locus.” Suggest change to “…dominant causal locus of decreased rice grain length.”.

Author Response

We would like to express our sincere gratitude for reviewing our paper and providing valuable feedback. We truly appreciate the comments and suggestions you have made. Please find below our responses and revisions addressing your concerns. We hope that the explanation has fully addressed all of your concerns.

  1. Comments and suggestions on English editing and writing

Response: We have adopted and corrected all the suggestions and comments on English editing and writing in this paper.

  1. Line 46) “…exhibited a more severe phenotype, whereas another showed a mild phenotype.” More severe or mild than what? The wild type?

Response: Thank you for your great suggestion on improving the readability of our sentences in the paper. Both the more severe and milder phenotypes described here were compared with wild-type RGA1 allele. We have clarified this in the new manuscript.

  1. Line 129) “…which were grown for two weeks…”. How were these plants grown? Perhaps this is described elsewhere but should be specified since mRNA profiles will change with different growing methods.

Response: Thank you for your comments on the culture environment of transgenic line and their control group in our article. The materials used for qPCR experiments were cultured on photoculture rack in the greenhouse. We give the plants a long-day conditions and control the ambient temperature of 28 ℃, relative humidity of 70% and light intensity of 15,000 Lux. We have supplemented this in the new manuscript.

  1. Line 151) “…was roughly 3:1 in F2samples…”. The numerical results should be reported and a non-parametric statistical test (chi-square) of the 3:1 ratio confirmed.

Response: Thank you for pointing out our statistical omissions in sample ratio. We analyzed the grain size phenotypes of 188 F2 generation individuals in field environment. Among them, 146 strains showed a large grain phenotype similar to Quan9311A, while 42 strains showed a small grain phenotype similar to Fuhui212 (Chi-square = 0.70922, df = 1, P = 0.3997). We added this information to the manuscript.

  1. Lines 180-181) “We phased the genotypes of each marker based on the genotype of Quan9311A.” The reviewer is confused about what the authors mean by “phased”. More explanation would be helpful.

Response: Thank you for your comments on genotype phasing, which made us realize that our explanation of the analysis process was not clear enough. We mapping the re-sequencing data of two parents, Quan9311A and Fuhui212, and their F2 individuals to the reference genome to obtain the SNPs and INDELs distribution of each sample, and used them as markers. For each marker, the parental source of every F2 sample (i.e., homozygous in Quan9311A-type, heterozygous, or homozygous in Fuhui212-type) is assigned by comparing with the genotype of Quan9311A. In this way, we can distinguish which genes in F2 individuals are from Quan9311A, which genes are from Fuhui212, and which genes are in a heterozygous state. We have added these details in the revised manuscript.

  1. Lines 201-202) “Over 95.9% of the co-segregated variants were predicted to have no large impact on coding regions…” the reviewer is confused by this assertion. More explanation is needed; what impacts are inferred?

Response: Thank you for your comments on our data analysis process. We used the snpEff software predict the effect of each variant based on the gene annotations of the reference genome to see whether any variant might cause functional alterations in these protein-coding genes. Some mutations with significant functional effects are the ones we focus on, such as frameshift mutations, premature termination, splicing site changes, and etc. We have supplemented this in the new manuscript.